# Separate and unequal: Moral domains differ in corresponding social judgments of others

Savannah Adams👁1☉*, Oscar Ybarra²☉

1 Department of Psychology, University of Michigan, Ann Arbor, Michigan, United States of America,
2 Gies College of Business, Champaign, Illinois, United States of America

☉ These authors contributed equally to this work.
* snladams@umich.edu

## Abstract

Current research on morality supports the idea that the moral landscape is comprised of several domains. However, the extent to which these domains may be thought of as equivalent when used as the basis for forming impressions or making social judgments is not yet understood. Past literature suggests that there may be notable differences in the evolutionary and social development of different moral behaviors, which raises questions about how actions based in different domains may be interpreted and judged by others. Across three studies, we had participants evaluate social targets based on behaviors pertaining to the moral domains. Results showed that correspondent inferences, attributional bias, and willingness to cooperate differed across domains. Interestingly, the *Equality* and *Property* domains emerged distinct. This research contributes to contemporary morality literature and proposes a new direction for understanding how morality plays a role in social judgment.

## Introduction

If you were to open a news app, you would likely see a litany of stories that detail a variety of moral or immoral acts. For instance, you may see news of internet scams aimed toward stealing money from individuals with healthy nest eggs, or people denied basic health and legal services due to their gender. Other news stories might describe how a man courageously fought off a wild animal to help his dog or how a small startup developed a new way to help their community fight off a water crisis. Different days bring different moral tales, but over time similar themes of morality repeat. What do we do with this social information? Do we weigh these moral acts the same, or are some more potent in the impressions they leave behind?

Contemporary work on moral cognition has mapped out the moral landscape to suggest distinct domains of social behavior related broadly to cooperation and the regulation of relationships. Still, it remains unclear whether we attach the same gravity to acts within different domains. The present research investigates the weight

**Data availability statement:** All primary data and analysis scripts are publicly available (https://doi.org/10.17605/OSF.IO/P5ZCX).

**Funding:** The author(s) received no specific funding for this work.

**Competing interests:** The authors have declared that no competing interests exist.

given to various moral acts and how social judgments may differ as a function of moral domain.

## Morality as a set of domains

Evolutionary theories of the origins of morality shifted the early landscape of moral research, expanding upon theories that conceptualized morality as foundational for social order [1] by rooting altruistic behaviors in kin selection and reciprocity [2,3]. These theories encouraged framing morality not just as an abstract social concept, but as a set of evolved behavioral tendencies [1,4].

Moral researchers across the field have grouped these tendencies into various key domains encapsulating specific moral emotions, motivations, and behaviors thought to encompass the breadth of morality [1]. For instance, Shweder and colleagues [5] theorized that morality may be divided into three domains that capture behaviors concerned with harm and fairness (autonomy), catering to one's group (community), and divinity (purity). These were later expanded into Moral Foundations Theory (MFT; [6]), which currently suggests the presence of six major moral domains: doing care or harm, treating others equally, giving to others in proportion to their actions and contributions, staying loyal to one's ingroup, respecting authority, and purity or spirituality.

Of particular interest to the current research are cooperation-based [7] and relational [8] models, which highlight the function of moral behaviors within social communities and relationships. Curry's Morality as Cooperation model (MAC; [7]) ties moral behavior to domains evolutionarily relevant to increasing mutual cooperation and include domains such as helping kin, reciprocating actions, deference to authority, fair division of resources, respect for property, and bravery. Rai and Fiske's [8] model of morality in relationship regulation identifies four moral motives prominent in the maintenance of relationships and the judgment of others: unity (the motive to support one's ingroups), hierarchy (maintenance of a social hierarchy), equality (reciprocity and equal treatment), and proportionality (rewards matching merit and the balancing of costs and benefits).

Despite some differences in the explicit domain makeup of these models, they all have a common theme of morality as a way of ensuring mutual benefits for individuals within social groups and relationships [8,9]. To receive the benefits of living within social communities, one must present themselves as an individual worthy of receiving them [9]. Violating the relational and cooperative expectations of society, such as through the act of taking too much for oneself or acting outside of group interests, could potentially incur penalties such as social rejection and group exclusion.

## Morality and social judgment

Much previous work has established the broad importance of moral information in forming and maintaining social impressions. While research on impression formation has traditionally been dominated by concepts of communion (warmth) and agency (competence) [10], the construct of communion is often used interchangeably with morality or morality is conceptualized as one component of it [11,12]. However, when morally relevant character traits (e.g., kindness, generosity) and communion traits

less related to morality (e.g., sociability, agreeableness) are compared, morally relevant traits are granted more weight in impression formation [13]. These morally relevant traits also seem to be more highly prioritized and more quickly processed than traits related to agency [14–16]. These findings suggest a unique importance of morality in making social judgments.

But, although often thought of in social cognition literature as a monolithic category, the breaking up of moral acts into various domains raises an interesting question with regard to how these domains may factor into social impressions and character judgments. Some existing moral literature implies that certain moral domains may be more culturally utilized or that some moral domains may be flagged by others as indicative of inherent character flaws [17], and research by Everett and colleagues [18] have suggested moral actions adhering to deontological rules may lend themselves to more favorable person perceptions. These bodies of work suggest that some types of moral actions may be considered differently when forming impressions of others' moral character.

Despite these arguments, there is very little work to date that investigates whether actions in line with specific moral domains may differentially lend themselves to social impressions and cooperation. For example, we do not know whether adhering to the *Equality* domain results in stronger social inferences than adhering to the *Family/kin* domain, or whether a violation in the *Reciprocity* domain results in stronger negative inferences than, say, violations related to the *Hierarchy* domain. Additionally, even less work explores this question within the scope of everyday behaviors; instead, much contemporary moral research relies on stimuli depicting situations that are very extreme (e.g., murder) or unlikely to exist in someone's daily life (e.g., the trolley problem) [19,20].

Understanding whether certain moral domains have more sway than others in our everyday social inferences may help to better understand the decisions and social interactions that result from those inferences. In the present research, we aimed to gather an empirical base of moral domains and assess whether judgments of others' moral character differed as a function of these domains.

To do this, we synthesized a list of moral domains based on existing literature and utilized measures of correspondent inferences [21] to capture judgments of social targets following behaviors in line with and in violation of these domains. In addition to this, we also included measures of dispositional and situational attribution to explore whether behaviors in some domains are more likely to be attributed to someone's character than to external circumstance. Research has suggested that whether an individual perceives someone else's behavior to be driven by their character or a situational circumstance may play a role in intuitive moral character judgments and shape beliefs about the kind of person someone is [22,23]. In particular, dispositionally attributed behaviors are likely to be perceived as in line with someone's personal beliefs and identity. We aimed to measure whether acts relevant to some moral domains may be more likely to be attributed to one's character than acts in other domains. Some asymmetry in attribution following immoral acts has been suggested in past research, [24], though this research has primarily focused on how individuals may differently perceive moral norm violations that harm others vs. those that violate norms of purity. In our research, we aimed to investigate potential asymmetry between a larger set of proposed moral domains suggested by contemporary literature without focusing specifically on moral violations.

In Study 2, we included judgments of willingness to cooperate with social targets in line with definitions of morality as essential for inducing cooperation and regulating relationships [8,9]. In Study 3, we further tested the robustness of our discoveries—that some domains are given more weight than others in social judgment—by assessing whether cognitive load could disrupt these inferences.

## Study 1

In Study 1, we investigated two questions: whether moral domains lend themselves differently to the inferences one makes about an individual, and whether actions pertaining to some domains might be more readily attributed to an individual's character rather than to external circumstance. We suspected that inferences made about another's character

following domain-specific behaviors, referred to in literature as correspondent inferences [21], might highlight variations in how heavily these domains are considered when judging others. Also of interest was whether some moral domains are more likely to result in dispositional rather than situational attributions, regardless of whether an act is in line with a moral domain or in violation of it (e.g., to steal is because you are a bad person and to not steal is because you are a good person). This study was largely exploratory to ascertain if and how social judgments may differ as a function of moral domain.

## Method

We recruited $N = 60$ U.S. participants on Prolific Academic ($M_{Age} = 32.9$, $SD_{Age} = 11.52$, 83.3% Female). We wanted to ascertain broadly whether differences exist across moral domains, but we did not have specific hypotheses about what domains any variation should stem from. We aimed for a sample large enough to detect small to medium variation across our domains ($d = .3$) within our repeated measures design. Using G*Power, we ran a power analysis for a fully within-subjects repeated measures ANOVA. At 80% power, we would require a sample of $N = 30$ to detect our desired effect [25].

### Procedure

Participants provided judgments of 14 distinct individual persons based on behaviors these individuals enacted, either in a way consistent with a moral domain (positive behaviors) or in violation of it (negative behaviors). Participants were asked to provide impressions of these individuals (targets) based on their behaviors. Targets and associated behavior descriptions were shown in random order. Following each target-behavior description, participants indicated their inferences about the target and attributions for their behavior. A cognition check was included at the end asking participants to create three distinct English words using the letters "TNKIH".

### Materials

**Moral domains.** The current studies used a set of moral domains inspired by those included in Curry, Chesters, and Van Lissa's [7] MAC model, as well as Rai and Fiske's [8] Relational Model. Initially, this gave us the following 11 domains as a starting point for our research: behaviors relating to *Family, Group, Reciprocity, Bravery, Deference, Fairness, Equality, Property, Proportionality,* and *Unity/Communal Sharing*. Our use of these models for extracting our base domains was due to multiple factors. First, these models together incorporate many of the themes and domains that also exist in other models of moral behavior (e.g., MFT) [9]. Second, these domains have been specifically tied to interpersonal interaction and relational prescriptions [8,9].

While the domains within these models formed the starting point for our own domains, we wanted to avoid using them as the sole lens for understanding how morality may be commonly understood and experienced. Instead, we aimed to construct a bottom-up view of morality by encouraging participants to generate their own examples of behaviors that they believed may align with or violate these domains. Starting with the definitions of the domains offered by Curry et al., [7] and Rai and Fiske [8], participants in a series of pilot tests were asked to come up with real behaviors (based on their own lives or others') that encompassed each of the domains.

These participant-generated behaviors were then distributed to new samples, who were asked to recategorize each of them into the provided domains and assess how favorable (positive or negative) they were. Conceptual overlap among our behaviors led to the consolidation or omission of certain domains due to content similarity or participants' overclassification of behaviors into those domains (e.g., consolidating the *Hierarchy/Deference* domains or removing the *Group* domain due to an overproportion of behaviors being assigned to it). Our final domains consisted of the seven included in this research and definitions of each were created based on the overall themes of the participant-generated behaviors within them. While each domain was created based on multiple behaviors provided by participants, in the current research we focused on using the constructed definitions for each domain rather than the individual behaviors from participants.

This was meant to allow us to test for potential differences in judgment across these domains without the confounding effects of contextual cues. We believed this approach would be ideal for an initial investigation into the potential differences between moral domains.

We adapted the behavioral definitions for our moral domains into pairs of behaviors belonging to each domain: one positive behavior (consistent with the domain) and one negative behavior (in violation of the domain). This provided us with 14 behavior statements in total, and each was associated with a different four-letter male name. For example, "Mark helped a member of his family" or "Dave failed to help a member of his family". Our list of adapted stimuli can be found in Table 1.

**Correspondent inferences.** The measure of correspondent inferences was inspired by Gilbert et al.'s [26] work on inferences pertaining to trait anxiety. We adapted the items from his study to center on morality, based on the Morality as Cooperation literature [7,9]. Participants were asked whether they felt that the social targets carrying out each behavior were principled, ethical, and morally upstanding on a 6-point bipolar scale (e.g., 1 = *Unprincipled* and 6 = *Principled*) ($a$ = .92−.97). A bipolar measure of likability (i.e., "I think that this person is…", 1 = *Unlikable* and 6 = *Likable*) was also gathered but was considered to thematically differ from the other inferences and was left out of the final scoring. Responses to this item echoed those of the other inference measures, such that behaviors in the *Equality* and *Property* domains elicited more extreme responses, $F(6, 354) = 21.98$, $p < .001$, $\eta^2_p = .27$, $d = 1.22$.

**Attribution.** The measure of attribution was adapted from Fincham and Bradbury [27]. The original measure only tested causal attributions to the person, so we added an item to determine situational attribution. Attributional items were "This person's behavior was due to something about him (e.g., the type of person they are, the mood they were in)", "This

**Table 1. Stimuli for Domains in Study 1.**

| Family | |
|---|---|
| Positive | Mark helped a member of his family. |
| Negative | Dave failed to help a member of his family. |
| Reciprocity | |
| Positive | Kyle did for someone else as they had done for him. |
| Negative | Joey failed to do for someone else as that person had done for him. |
| Bravery | |
| Positive | Pete behaved in a way that is courageous, tough, or resilient. |
| Negative | Luke failed to behave in a way that was courageous, tough, or resilient. |
| Hierarchy | |
| Positive | Phil honored the rules of his superiors. |
| Negative | Jake failed to honor the rules of his superiors. |
| Equality | |
| Positive | Theo showed equal kindness and compassion to all others. |
| Negative | Bill failed to show equal kindness and compassion to all others. |
| Property | |
| Positive | Paul respected someone else's property. |
| Negative | Alan failed to respect someone else's property. |
| Unity & Communal Sharing | |
| Positive | Drew united with his community to share burden and responsibility. |
| Negative | Matt failed to unite with his community to share burden and responsibility. |

person's behavior was due to something about the situation (e.g., his environment, some external pressure)", and "This person's behavior is representative of the kind of person they are". Participants responded on a scale from 1 = *Strongly Disagree* to 6 = *Strongly Agree.* The item asking if the target's behavior was about them and the item asking if their behavior is representative of the kind of person they are strongly correlated, $r_{rm}(778)$ =.56, $p<.001$, so we combined these into a single measure of dispositional attribution.

Despite having no specific expectations of which domains might produce distinct social judgments, we expected that any variation in how individuals use moral domains to judge social targets would manifest through more extreme correspondent inferences and dispositional judgments about those targets. That is, individuals who learn of a social target acting consistently with or violating a moral domain should rate that target as more extremely moral or immoral, respectively.

### Difference scores

Two kinds of difference scores were calculated to assess variations across domains. The first provided an overall sense of the correspondent inferences made about social targets. Correspondent inference scores following negative behaviors (i.e., domain violations) were subtracted from those following positive behaviors. Higher difference scores indicated greater polarization of the inferences associated with behaviors in each domain (e.g., more positive judgments of positive behaviors and more negative judgments of negative behaviors).

The second set of difference scores pertained to attribution. Situation attribution ratings were subtracted from disposition attribution ratings. Higher difference scores indicated a greater overall dispositional judgment of behaviors within the corresponding domain. Importantly, attribution difference scores were obtained for both positive and negative behavioral stimuli in each domain, providing 14 overall difference scores. This contrasts with the correspondent inference scores, which, due to subtracting scores for negative behaviors from those for positive behaviors, only resulted in one difference score per domain.

While researchers are aware of the debate surrounding the use of difference scores in research [28], employing difference scores in the present analyses allowed for clearer emphasis of the differences in judgment patterns across the domains. To ensure that there were no emerging patterns that were driven by judgments as a response to behavior valence (i.e., stronger judgments following negative or positive behaviors resulting in larger difference scores for a domain), we also conducted analyses without using difference scores. These showed no patterns suggesting any of our effects were driven solely by behavior valence. These elaborated results not using difference scores can be found in S1, S3, and S5 Appendix in the Supporting Information.

### Results

A repeated measures ANOVA on the correspondent inference scores revealed a main effect of moral domain, $F(6, 354)$ = 20.59, $p<.001$, $\eta^2_p$=.26, $d$=1.18, such that inferences, regardless of behavior valence, were more extreme in some domains than others. Fig 1 shows a graph of these difference scores. Larger values in the *Equality* and *Property* domains indicate that ratings of others' moral character (i.e., whether they are ethical, principled, and morally upstanding) were higher following positive behaviors and lower following negative behaviors. Bonferroni-corrected pairwise comparisons confirmed the *Equality* and *Property* domains were statistically different from all other domains but not from each other. This suggests these domains lend themselves to stronger correspondent inferences overall.

A repeated measures ANOVA on the attribution difference scores also revealed a main effect of domain, $F(5.2, 301.53)$ = 24.39, $p<.001$, $\eta^2_p$=.3, $d$=1.32. Bonferroni-corrected pairwise comparisons showed that participants were more likely to make dispositional attributions for behaviors corresponding to *Equality* and *Property* than all other domains. The Bonferroni-corrected multiple comparisons tests for Study 1 are reported in S2 Appendix in the Supporting Information.

The analysis also produced a main effect of valence, $F(1, 58)$ = 20.52, $p<.001$, $\eta^2_p$=.26, $d$=1.18, and an interaction of domain and valence, $F(6, 348)$ = 3.7, $p<.001$, $\eta^2_p$=.06, $d$=.51.

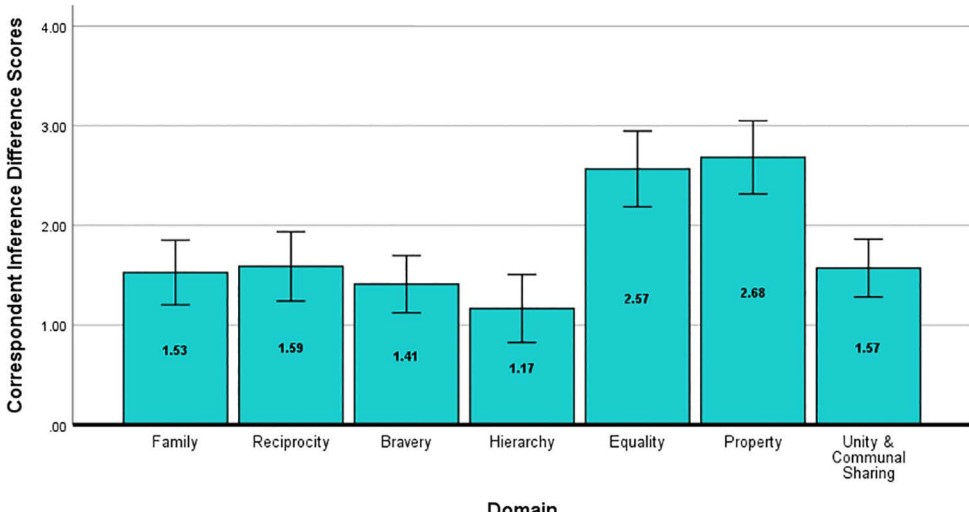

**Fig 1. Correspondent Inference Difference Scores for Study 1.** Difference scores for correspondent inferences about a social target (i.e., how ethical, principled, and morally upstanding they are) following positive and negative behaviors in each moral domain (positive – negative). Higher values indicate more extreme judgments in a domain (more positive following behaviors in line with a domain and more negative following domain violations). Error bars represent the 95% confidence interval.

The valence main effect indicated that participants were more likely to attribute behaviors to a social target following a positive ($M = .64$, $SD = 1.28$) than a negative behavior ($M = .21$, $SD = 1.42$). The interaction indicates the difference in judgments following positive and negative behaviors was more pronounced in some domains than others. As visualized in Fig 2, the findings suggest a distinction for the *Equality* and *Property* domains in the formation of social judgments.

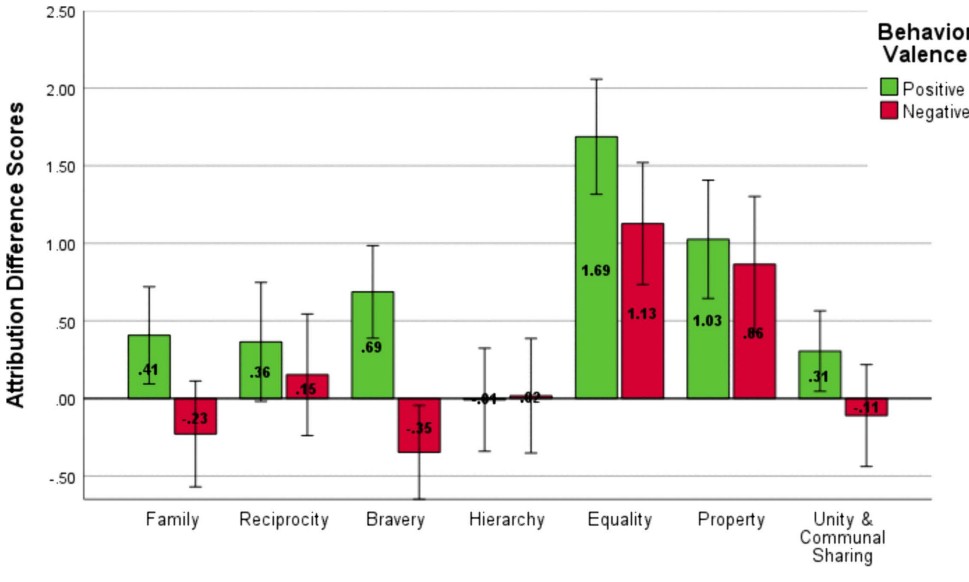

**Fig 2. Attribution Difference Scores for Positive and Negative Behaviors Across Domains in Study 1.** Difference scores for attribution (dispositional attribution – situational attribution). Higher values indicate a higher overall dispositional judgment following a behavior in a domain. Error bars represent the 95% confidence interval.

Given that this sample was 83.3% female, we wanted to make sure that there was no easily identifiable effect of gender on the results. While the power for adding a between-subjects variable to check for main effects was low, we ran a separate analysis with gender included to check for any main effects. Gender was shown to have a nonsignificant effect on both the correspondent inferences about a social target ($p = .32$) and any attributional judgments ($p = .07$).

### Supplemental analyses

To ensure that the statistical differences observed in the *Equality* and *Property* domains were not due to the behaviors within these domains being more or less aversive than others, we conducted a study ($N = 27$) to ascertain the perceived valence of each behavior. Participants were shown behavior descriptions that did not name a social target (e.g., "X failed to help a member of their family" instead of "Dave failed to help a member of his family") and were asked to rate how positive or negative each behavior was from 1 = *Very Negative* to 7 = *Very Positive*. While some differences in favorability across domains and conditions were found, the patterns found cannot account for the results of Study 1 (see S1 Fig in the Supporting Information).

We collected a second sample of participants ($N = 57$) to further confirm that variance of judgments based on *Equality* and *Property* behaviors were not due to peculiar qualities of the domains. The stimulus dimensions we assessed were: ease with which example behaviors come to mind, perceived confirmability (i.e., number of times behavior needs to be observed before attributing it to a person), frequency with which one acts in ways that correspond to each domain, frequency with which they would expect others to do the same, and whether individuals felt each domain was broad or narrow. No patterns emerged to suggest the distinct judgments following behaviors in the *Equality* and *Property* domains were prompted by peculiarities in the makeup of these domains. All materials and results as well as a full list of graphs and comparisons can be found in S1 Text in the Supporting Information.

## Study 1 Discussion

Study 1 provides initial support for the idea that some moral domains may differ in their contribution to social impressions of others. In particular, moral actions relating to *Equality* and *Property* led to more positive judgments of social targets' moral character following behaviors in line with these domains, and more negative judgments of targets' moral character following behaviors violating these domains. Actions within these domains also appeared more likely to be attributed to targets' disposition rather than to situational circumstances or environment. Despite some variation in the attributional features of the domains themselves (e.g., how favorable they are, how easy it is to think of behaviors relevant to each domain), these effects do not seem to explain the results of Study 1. There were also no gender effects that might explain the results of Study 1.

We can look to the developmental literature or distinct psychological phenomena such as the endowment effect [29] to suggest reasons to suspect why *Equality* and *Property* might behave differently than other moral domains in the context of social judgment. However, before speculating about why these domains should behave differently than other moral domains in the context of social judgment, we first wanted to replicate the results with elevated power for our paired comparisons.

## Study 2

Study 2 included both a larger sample and a measure of one's willingness to cooperate with a social target in line with Curry's [9] framing of morality as an evolutionary solution to problems of cooperation. We predicted that, consistent with the results from Study 1, there would be differences in how domain-specific behaviors influence participants' willingness to cooperate with social targets.

### Methods

Study 1 resulted in medium to large effect sizes for our repeated measures tests. Nevertheless, we wanted to ensure elevated power for our paired comparisons. G*Power [25] indicated that the required sample size to detect a $d = .3$ effect size

at 80% power within a paired t-test was $N = 90$. We recruited 120 participants on Prolific Academic. Fourteen participants were removed for failure to complete the cognition check, resulting in a final sample of $N = 104$ ($M_{Age} = 35.38$, $SD_{Age} = 12.21$, 62.7% Female).

Like Study 1, participants were presented with 14 total behaviors, seven positive and seven negative, associated with 14 different social targets. The same names for each target from Study 1 were used in Study 2. Following each behavior, participants were asked to indicate their inferences about the target, dispositional and situational attributions, and willingness to cooperate with the target. Behavior statements were presented in random order.

### Materials

**Moral domains, correspondent inferences, and attribution.** The behavioral stimuli as well as the correspondent inferences and attribution measures used were the same as in Study 1. The same cognition check was also used in Study 2.

**Cooperation.** After each behavioral statement (e.g., "Dave failed to help a member of his family") and following the correspondent inference and attribution measures, participants indicated the extent to which they would be willing to tell the social target, whom they did not know, a personal secret, trust them to solve a dispute, seek advice from them, or share a car or ride with them. These measures were adapted from Wojciszke et al. [15], though the item about sharing a car or ride was added to provide an additional measure of approach-avoidance. Participants indicated their willingness on a 6-point scale from 1 = *Extremely Unwilling* to 6 = *Extremely Willing* ($a = .80 - .95$).

### Results

As done for Study 1, we created difference scores for correspondent inferences and attribution. In Study 2, we also created a set of difference scores for participants' willingness to cooperate with social targets. These were obtained by subtracting scores of cooperation following negative domain-specific behaviors from those following positive domain-specific behaviors. Higher difference scores indicate greater willingness to cooperate with targets following positive behaviors and less willingness to cooperate following negative behaviors.

Repeated measures ANOVAs replicated the results of Study 1 for correspondent inferences. We observed a significant effect of domain, $F(5.57, 573.93) = 20.9$, $p < .001$, $\eta^2_p = .17$, $d = .90$. As depicted in Fig 3, Bonferroni pairwise comparisons indicated there were larger differences in judgment following positive and negative behaviors in the *Equality* and *Property* domains. These domains were statistically different from other domains but not from each other. All Bonferroni-corrected multiple comparisons tests for Study 2 are reported in S4 Appendix in the Supporting Information.

Variations in attributions following moral behaviors were also replicated in Study 2. Most important was the main effect of domain, $F(5.6, 576.68) = 36.66$, $p < .001$, $\eta^2_p = .26$, $d = 1.18$, which indicated that among the moral domains, *Equality* and *Property* were most likely to be judged as dispositional (see Fig 4). A main effect of valence was also found, $F(1, 103) = 35.87$, $p < .001$, $\eta^2_p = .26$, $d = 1.18$; positive behaviors were perceived as driven more by targets' dispositions ($M = .64$, $SD = 1.28$) than negative behaviors ($M = .21$, $SD = 1.42$). An interaction between domain and valence indicated variation in the magnitude of this difference across moral domains, $F(5.35, 550.6) = 5.51$, $p < .001$, $\eta^2_p = .05$, $d = .46$.

Novel to Study 2 was our investigation of willingness to cooperate with a social target based on behavioral information pertaining to moral domains. The repeated measures ANOVA revealed a main effect of domain, $F(5.26, 541.55) = 24.64$, $p < .001$, $\eta^2_p = .19$, $d = .96$. These difference scores are shown in Fig 5. The difference scores for behaviors in the *Equality* and *Property* domains were larger, indicating that positive behaviors in these domains led to a greater willingness to cooperate with social targets (i.e., share secrets, seek advice, trust with solving disputes, share a car), while negative behaviors did the opposite. This supports our hypothesis that willingness to cooperate would differ as a function of moral domain in line with correspondent inferences and attribution, as well as the results of Study 1. There were no statistical differences between responses based on gender in any of the main analyses for Study 2.

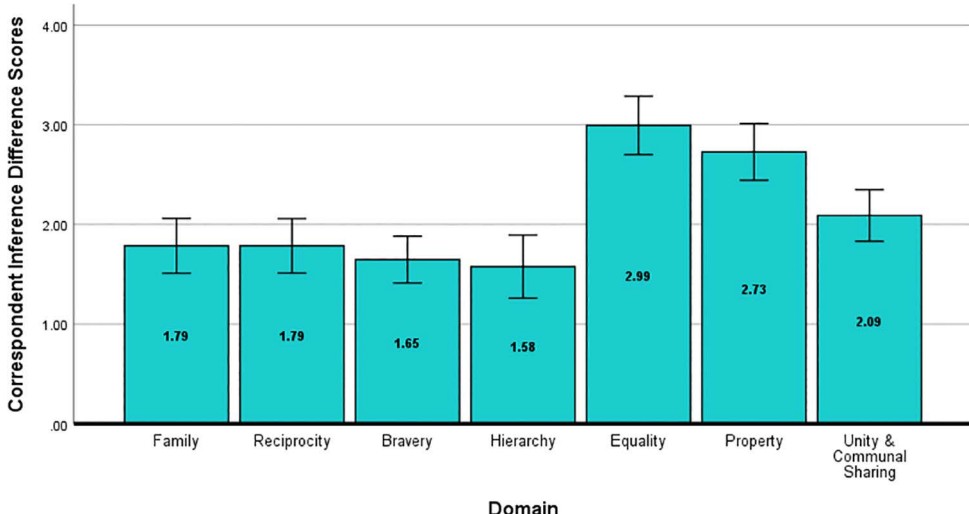

**Fig 3. Correspondent Inference Difference Scores for Study 2.** Difference scores for correspondent inferences about a social target (i.e., how ethical, principled, and morally upstanding they are) following positive and negative behaviors in each moral domain (positive – negative). Error bars represent the 95% confidence interval.

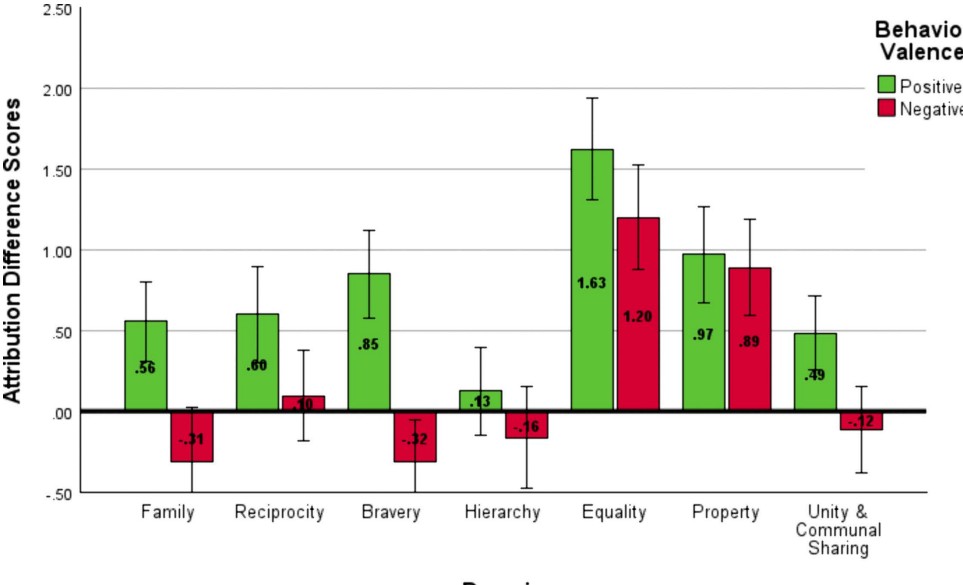

**Fig 4. Attribution Difference Scores for Study 2.** Difference scores for attribution (dispositional attribution – situational attribution). Higher values indicate a higher dispositional judgment.

## Study 2 Discussion

Study 2 replicated the findings from Study 1 with participants making more extreme inferences about targets' character and more dispositional attributions following behaviors related to the *Equality* and *Property* domains. This study also extended these findings by showing that participants were also less willing to cooperate with social others in ways

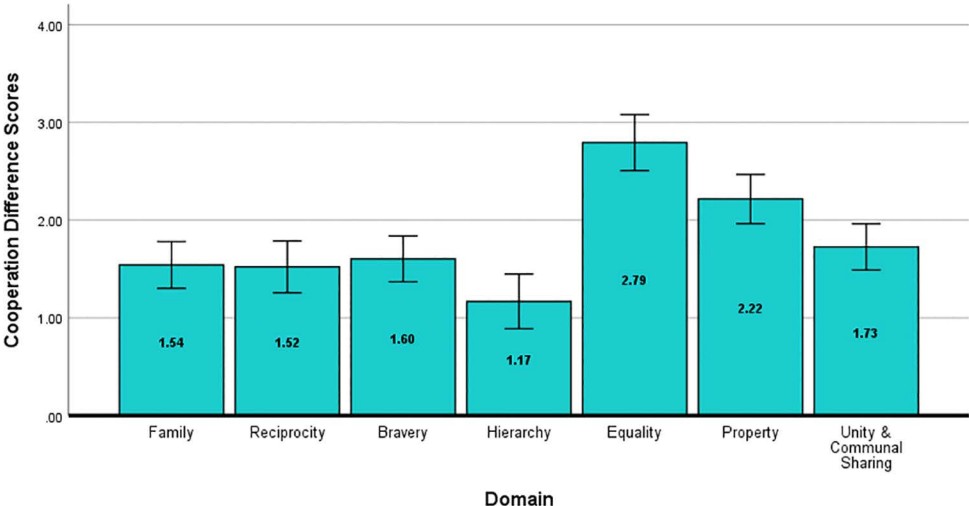

**Fig 5. Cooperation Difference Scores for Study 2.** Difference scores for participants' willingness to cooperate with a social target (i.e., share a personal secret, seek advice, trust with solving a dispute, share a car or ride) following behaviors in each moral domain (positive – negative). Error bars represent the 95% confidence interval.

like sharing a car, telling them personal information, or trusting them to solve a dispute following behaviors in these domains.

The robustness of these findings in a higher-powered study suggests that these domains have unique influence in our perceptions of others' character and the initial impressions we form about them. We speculate about possible reasons why these domains might emerge distinct in judgments in the General Discussion.

## Study 3

Studies 1 and 2 indicate differences in how individuals form correspondent and dispositional judgments of others based on behavioral information from different moral domains. In addition to magnitude, though, a key way social judgment processes reflect robustness is in how well they tolerate divided attention [30]. Thus, we conducted a pre-registered study in which we administered a digit task to induce cognitive load in line with prior research suggesting that this paradigm may be efficient in inducing cognitive busyness [31]. If behaviors relating to *Equality* and *Property* are processed more automatically or intuitively, they should be less disrupted by cognitive tasks that inhibit deliberate processing [26]. Based on the growing evidence that *Equality* and *Property* domains are treated preferentially in social judgment, we expected that that would continue to be the case even when participants performed the social judgment task under cognitive load.

## Method

Study 3 included a mixed design inspired by Gilbert and Osborne's [31] digit-span task. Participants were randomly assigned to between-subjects conditions in which they were shown the same repeated measures stimuli from Studies 1 and 2, but with the addition of having to memorize either a two-digit or eight-digit number when reading each vignette and accurately record it after answering each set of outcome items.

Initially, our preregistration for this experiment stated that participants in one condition would see an eight-digit number while the other saw only the moral stimuli without the addition of a memorization task. We chose to instead present this other group with a two-digit number to memorize so that we could ensure more consistency between groups and more directly investigate differences in perceived cognitive effort between the conditions.

In order to account for the random assignment of participants to groups in Study 3, we recruited 160 participants. Our preregistration specified that we would aim for approximately 200 participants, but a power calculation for a mixed repeated measures analysis with a single between-subjects variable indicated we would need a sample of $N = 82$ to detect a medium effect of $d = .5$ at 80% power with moderate correlation between the variables [25]. Some participants did not record any numbers from the digit task or failed the cognition check, leaving a final sample of $N = 142$ ($M_{Age} = 38.13$, $SD_{Age} = 12.34$, 51.4% Male).

Participants were told at the beginning of the survey that they would be assigned to one of two groups in which they would be given a digit memorization task. This task would require them to remember a series of either two-digit or eight-digit numbers while making judgments of other people. Similar to the instructions used in Gilbert and Osborne [31], participants were told that the purpose of this task was to test how skilled people are at doing multiple tasks simultaneously. Before each behavior was shown to the participant, they were given 15 seconds to memorize a two-digit or eight-digit number. After reading the behavior and making judgments about the target, they were asked to enter the number they were shown into a text box before moving on to the next number-behavior combination. Given that the survey was taken online, all participants were asked not to record or write down the numbers that they were shown and to only hold them in memory.

Study 3 used the same behavioral stimuli, correspondent inference measures, and attribution measures, and cognition check used in Studies 1 and 2. However, two items were added to the survey to gauge whether participants in the eight-digit condition were more challenged cognitively than those in the two-digit condition. These items asked participants to rate their agreement with the statements "It was difficult to keep each number in mind while assessing each individual's character." And "It took little effort to complete the number memorization task and the person judgment task at the same time." on a 7-point scale (1 = *Strongly disagree* to 7 = *Strongly agree*). The latter item was reverse-coded so that a higher scale number indicated greater effort.

## Results

Independent sample t-tests were conducted on the manipulation check items to ensure amount of effort and difficulty differed by condition. Participants in the eight-digit condition ($M = 5.54$, $SD = 1.35$) reported having greater difficulty completing the tasks than those in the two-digit condition ($M = 3.69$, $SD = 1.72$), *Welch's* $t(132.53) = –7.11$, $p < .001$, and that more effort was required ($M = 5.31$, $SD = 1.63$) to do both tasks than those in the two-digit condition ($M = 3.75$, $SD = 1.68$), $t(140) = –5.64$, $p < .001$.

As in Studies 1 and 2, we created difference scores of the correspondent inferences and attributions for each behavior and analyzed these with repeated measures mixed ANOVAs. Correspondent inference difference scores revealed a significant main effect of domain on the moral character judgments of each target, $F(6, 840) = 23.21$, $p < .001$, $\eta^2_p = .14$, $d = .8$, which are shown in Fig 6. Bonferroni-corrected pairwise comparisons showed identical patterns to previous studies: the *Property* and *Equality* domains emerged distinct from all other domains but not from each other. Importantly, there was neither a main effect of condition on these judgments nor an interaction effect, indicating that the inferences participants were making about each target were robust to the addition of cognitive load. The Bonferroni-corrected multiple comparisons tests for Study 3 are reported in S6 Appendix in the Supporting Information.

Regarding attributional judgments, the results again replicated the patterns found in Studies 1 and 2. As depicted in Fig 7, there was a main effect of moral domain on the attribution of behaviors, $F(5.47, 766.15) = 27.60$, $p < .001$, $\eta^2_p = .17$, $d = .9$, such that the *Equality* and *Property* domains were more likely to be dispositionally attributed than behaviors in other domains. There was also a main effect of valence, $F(1, 140) = 39.10$, $p < .001$, $\eta^2_p = .22$, $d = .1.06$; positive behaviors were more likely to be dispositionally attributed than negative behaviors at a mean difference of .56 (95% CI [.38,.73]). An interaction between domain and valence again showed variance in the magnitude of these effects across domains, $F(5.78, 809.51) = 4.81$, $p < .001$, $\eta^2_p = .03$, $d = .34$. No interactions were found between condition and moral domain or behavior

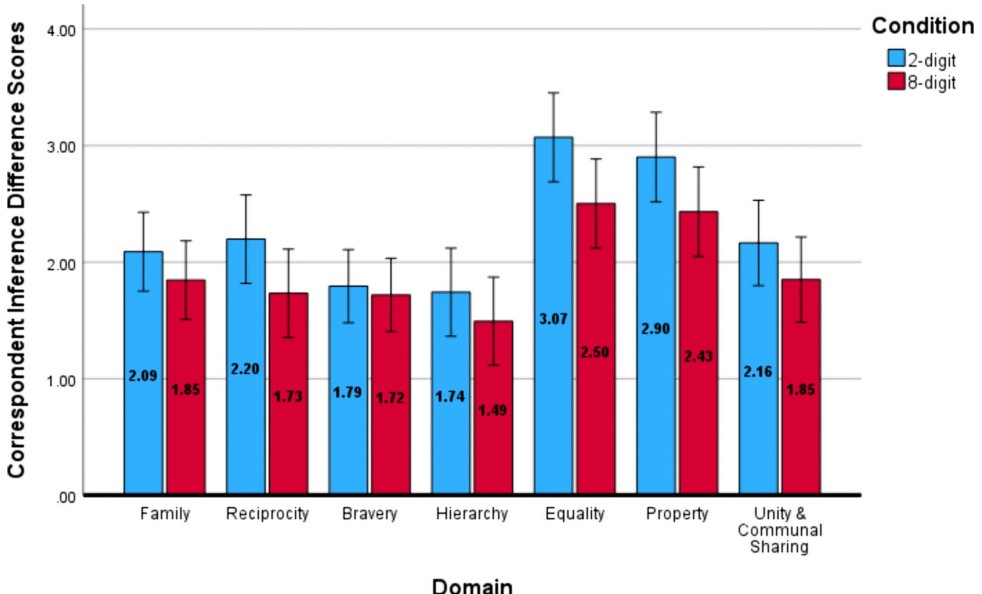

**Fig 6. Correspondent Inference Difference Scores for Study 3.** Difference scores for correspondent inferences about a social target (i.e., how ethical, principled, and morally upstanding they are) following positive and negative behaviors in each moral domain (positive – negative). Higher values represent more extreme judgments (positive judgments following positive behaviors and negative judgments following negative behaviors) in a domain. Error bars represent the 95% confidence interval.

valence, indicating that these patterns of attribution did not differ statistically dependent upon the digit tasks participants were given. This provides more support for the idea that these judgments may be consistent even under cognitive burden. There were again no statistical differences between responses based on gender in Study 3.

## General discussion

Studies 1 and 2 were conducted to investigate whether there were observable differences in the correspondent inferences and attributions individuals make about others following behaviors in line with different moral domains. Inferences and attributions did differ by domain; inferences about a target's moral character were stronger in the *Equality* and *Property* domains, and behaviors pertaining to these domains were also more likely to be attributed to a target's disposition. Study 2 included a measure of cooperation to ascertain whether an individual's willingness to cooperate with a social target differed across domains. *Equality* and *Property* once again emerged distinct, supporting Curry's [9] theory that moral behaviors developed to solve problems of cooperation in social communities. Study 3 demonstrated that the distinctiveness of correspondent inferences and attributions in the *Equality* and *Property* domains were robust to cognitive load, indicating inferences based on moral behaviors, at least within some domains, may be somewhat automatic and resistant to distraction.

Worth noting is that in all three studies, participants were more willing to attribute positive behaviors to social targets' character than negative behaviors. This runs counter to findings that negative information may play a larger role than positive information in impression formation (e.g., [14,15,32]). However, this may have been because participants were only presented one piece of information about each target and more readily made attributions about positive behaviors overall (this is supported by Fig B in S1 Appendix in the Supporting Information). Literature has suggested that positive moral behaviors are more strongly associated with social norms than negative behaviors [32], which may lead to general expectations of decency from others. This, paired with the fact that participants only saw a single positive and negative behavior

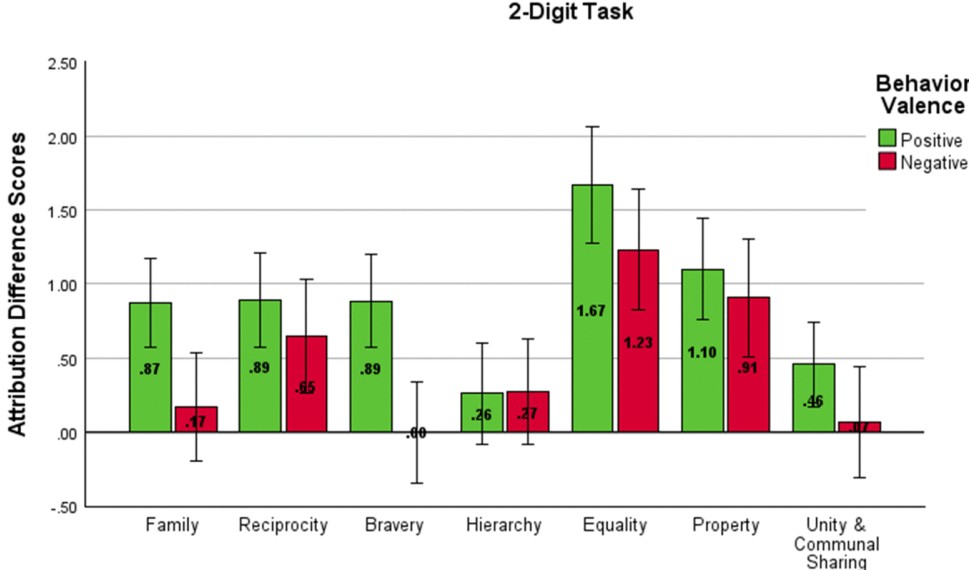

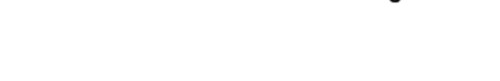

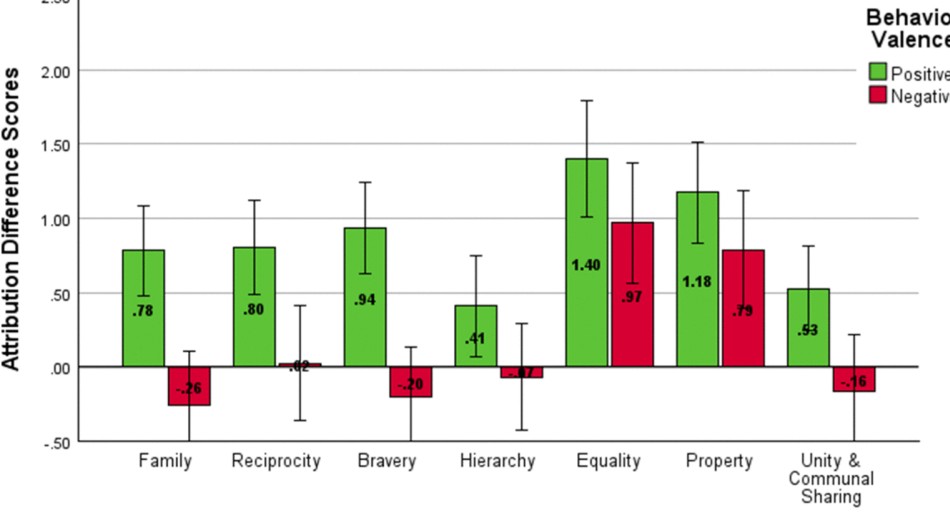

**Fig 7. Attribution Difference Scores for Study 3.** Difference scores for attribution (dispositional attribution – situational attribution). Higher values indicate a higher dispositional judgment.

for each domain, may have led to hesitance when making attributions and a tendency to default to positive inferences about social targets.

The distinctiveness of the *Equality* and *Property* domains in the current research raises questions about why these may lead to stronger inferences and dispositional attributions.

Developmental researchers have found that infants as young as 18 months old demonstrate a natural desire to treat others with equal kindness and engage in cooperative acts without regard for reward [33]. Additionally, some research has shown that children may make social judgments based on the helping behaviors they witness in others, and this may

impact their willingness to approach or avoid [34]. Engelmann and Tomasello [35] argue that children glean a sense of fairness from cooperative interactions and that this is used as a blueprint for resource distribution. By approximately three years old, children demonstrate an aversion toward inequity. These findings suggest a human tendency toward helping and showing kindness to others regardless of kin status or reciprocity, in line with the *Equality* domain.

The importance of property may be hinted at by observations inherent across many psychological phenomena. The endowment effect, through which individuals assign additional value to something that they perceive having ownership of, touches on the importance of property even without specific attention to its moral potential [29]. Research on children has also demonstrated a respect toward prior ownership and that ideas of ownership may be applied to concrete physical objects as well as more abstract items such as ideas and intellectual property [36,37]. Property law in the United States emphasizes concepts pertaining to prior possession and private ownership which may rely on some evolved sense of morality [38]. Merrill and Smith [39] argue that the systemic importance of right to property could not survive without moral relevance; it relies on collective beliefs among unconnected groups that interference with others' belongings is socially punishable. This ensures mutual cooperation toward this end.

While our current findings are insufficient for making concrete differentiations between each of the domains, these different literatures help paint a picture of *Equality*- and *Property*-relevant behaviors as related to early socialization and embedded within social systems of development and everyday living. Within an American context, it is possible that these behaviors have evolved to be more ubiquitous and consistently valued than behaviors in other domains, which may be in line with moral research suggesting cultural variation in moral values based on differences in social ecologies and traditions when implementing or teaching moral beliefs [17]. Indeed, while moral behaviors may overall be adaptive solutions to problems of cooperation and relationship regulation within communities [8,9], differences in how these moral behaviors manifest and their functions within social communities may influence how they are used in social impressions.

## Limitations and future directions

While our moral domains are based on contemporary social and relational moral models and most widely accepted moral domains made up the foundation of our stimuli, our process for narrowing the moral domains used in this work may potentially exclude domains that researchers have proposed to be relevant to moral judgments. For instance, we did not include domains such as *Liberty* or *Honor*, which have been seen as potential moral domains in MFT [40]. The exclusion of some moral domains in this work may limit the extent to which we may conclude that *Equality* and *Property* emerge distinct above other domains, however, our aim is primarily to show that there may be differences in the weight given to some types of moral actions over others when forming judgments and impressions of others.

Similarly, a potential limitation to this work, though a strength for our purposes in these studies, is that we used the definitions of our moral domains as generic behavioral stimuli in lieu of specific individual behaviors. While our stimuli definitions were created from individual participant-generated behaviors, using the definitions alone strips away context and potentially restricts the extent to which judgments made by participants might emulate those in real situations.

In the future, it will be important to consider how contextual information may impact how moral domains are considered when forming impressions and judgments of others. For example, it will be crucial to study whether a target person is known or is a stranger, as was the case in this research, and if known, the emotional closeness they share with the participant [41]. It is also important to note that the current set of studies was conducted on a sample from the United States and our findings may not generalize to other countries or cultures.

The current research is only a starting point for exploring variations in how moral domains affect judgment and behavior. For instance, how might these moral domains be weighted when it comes to building one's own personal reputation (i.e., is it better to be thought of as being a coward or a thief)? Ybarra et al. [42] found that individuals were more motivated to repair their reputations when they were thought to lack communion traits (e.g., honesty, kindness) rather

                                                                                                    

than agency traits (e.g., skill, success). It might be that individuals will be more motivated to repair their reputation when thought to be lacking in some moral domains over others.

## Conclusion

The present studies provide initial evidence that in the moral landscape, dimensions of social living are associated with different expectations and levels of force with which people judge others. This highlights several new directions for social researchers to pursue in trying to understand the evolution and impact of morality on social interaction. It also helps to add additional nuance and complexity to the perceived relationship between moral behavior and social relationships. Indeed, the role of morality in forming impressions and judgments of others may not be a simple matter of what is moral and immoral but may in fact be made up of domain-specific moral weightings.

## Supporting information

**S1 Appendix. Study 1 Analyses Not Using Difference Scores.**
(DOCX)

**S2 Appendix. Multiple Comparisons for Reported Analyses in Study 1.**
(DOCX)

**S3 Appendix. Study 2 Analyses Not Using Difference Scores.**
(DOCX)

**S4 Appendix. Multiple Comparisons for Reported Analyses in Study 2.**
(DOCX)

**S5 Appendix. Study 3 Analyses Not Using Difference Scores.**
(DOCX)

**S6 Appendix. Multiple Comparisons for Reported Analyses in Study 3.**
(DOCX)

**S7 Appendix. Means and Correlations Between Measures in Studies 1–3.**
(DOCX)

**S1 Fig. Favorability of Behaviors Across Domains.** Values represent the perceived favorability (positive or negative) of domain-relevant behaviors without attribution to a specific target. These judgments are of the behaviors themselves, rather than the social target. Higher values indicate a more positive perception of each behavior. Results indicated main effects of domain, ($F(3.81, 99.15) = 4.15$, $p < .001$, $\eta^2_p = .14$, $d = .80$) and behavior valence ($F(1, 26) = 157.17$, $p < .001$, $\eta^2_p = .86$, $d = 4.96$), with an interaction effect ($F(5.33, 138.50) = 6.62$, $p < .001$, $\eta^2_p = .20$, $d = 1$). However, while mean judgments of the favorability of negative *Property*-relevant behaviors did differ statistically from judgments of negative behaviors in other domains (excluding *Equality*), favorability judgments of *Equality*-relevant behaviors did not differ from any other domains. (TIF)

**S1 Text. Study 1 Supplemental Analyses – Domain Comparisons.**
(DOCX)

## Acknowledgments

The authors would like to thank the editors and anonymous peer reviewers who lent their expertise to provide constructive and helpful feedback.

## Author contributions

**Conceptualization:** Oscar Ybarra.

**Data curation:** Savannah Adams.

**Formal analysis:** Savannah Adams.

**Investigation:** Savannah Adams, Oscar Ybarra.

**Methodology:** Savannah Adams, Oscar Ybarra.

**Project administration:** Savannah Adams.

**Supervision:** Oscar Ybarra.

**Visualization:** Savannah Adams.

**Writing – original draft:** Savannah Adams.

**Writing – review & editing:** Savannah Adams, Oscar Ybarra.

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
