## [Decision Letter · Decision Letter 0]

1 Jul 2025

Dear Dr. Adams,

Thank you for submitting your manuscript to PLOS ONE. After careful consideration, we feel that it has merit but does not fully meet PLOS ONE’s publication criteria as it currently stands. Therefore, we invite you to submit a revised version of the manuscript that addresses the points raised during the review process.

**ACADEMIC EDITOR: **

Your work has merit, and I think the paper can be a valuable contribution to PLOS One. The main area for improvement is the theoretical foundation and embedding. This pertains to many aspects of the paper (see comments below), but first and foremost, the grounding of the research question. If you could help readers understand how different approaches speak to why domains might be more, less, or equally relevant for moral character judgments, this would add a lot of value.

We look forward to receiving your revised manuscript.

Kind regards,

Johannes Schwabe

Academic Editor

PLOS ONE

Journal Requirements:

Additional Editor Comments (if provided):

1. Theoretical Foundation: I agree with both reviewers that the theoretical foundation for your work needs improvement, and this applies to the very grounding of your work all across to final discussion. It is okay to not derive specific hypotheses, but your work does not exist in theoretical vacuum either. Relational models theory, moral foundations theory, and even research on moral identity can be drawn upon in the introduction and speaks to why certain domains might be more or less important in judging others moral character. For example, Haidt speaks about how violations in certain domains can disproportionately undermine global judgments because they signal "core flaws" (e.g., dishonesty). Also consider the work done on differential judgments of deontological versus utilitarian wrong- and right-doers (e.g., Jim Everett's work on this). Your research is currently presented without much of this context, which makes it hard for readers to understand the significance of your work and how it speaks to other findings.

2. I agree with reviewer 2 that the way you arrive at the domains studied needs to be more transparent. The methods underlying the domain selection are not clear enough, and we should learn a bit more about why two and why exactly those two frameworks were chosen to begin with.

3. The authors report the supplementary analysis as evidence that "the patterns cannot account for the results of Study 1". I tend to disagree. There are clear differences, and I feel like these results underline a central problem. Just because the positive and negative variants of a behavior create more polarized ratings on moral characters, why can we infer that the associated domain is more central for moral character judgments? Especially in combination with the fact that the behaviors were not sampled from a representative pool of all behavior potentially representing each domain, but arrived at in a fairly untransparent bottom-up approach, I do not see how we can really infer between-domain comparisons. This needs to be addressed more comprehensively.

4. The lack of theorizing and embedding the research in the literature leads to a few issues. For example, in the pre-text to Study 2, the hypothesis appears, that there would be differences in how domain-specific behaviors influence participants' willingness to cooperate with social target. This is apparently purely data-driven, but there is a wealth of literature that would speak to that topic. This also relates to the findings how positive behaviors were perceived as driven more by targets' dispositions than negative behaviors. This conflicts with other studies consistently reporting that violations are given more weight in character judgments. Such apparent contradictions need to be transparent and discussed comprehensively.

Minor comments

1. The power analysis of Study 1 is not well-founded. At this point, it is neither clear which exact effect you are interested in, which effect size would be reasonable to expect, neither which statistical test you are going to apply to test it. If this was just a convenience sample with no clear pre-specified hypothesis and power analysis, please just state that. If not, please describe the underlying assumptions more clearly.

2. The use of difference scores and the exact statistical method planned should be explained in the methods section, and not in the results.

3. The figures are not self-explanatory. The direction of the differences is not clear.

4. Please report tables detailing the Bonferroni-Tests in the supplementary material and refer to them when talking about their results.

5. When comparing means, report the means and their SD in-text, not just the mean difference and CIs.

6. Generally, please report the individual variables (positive and negative, actor versus situation attribution, etc.) in the supplementary materials, and potentially even replace the current bar charts by dumb bell plots or similar, depicting both individual values as well as the differences.

7. A clearer rationale for why we need the cognitive load paradigm is important would make it easier for readers to understand how Study 3 adds to the picture.

8. Please include standard means and correlations tables for all Studies, where possible based on individual variables, and add to the supplementary material if too clunky for the main manuscript.

Reviewers' comments:

Reviewer's Responses to Questions

**Comments to the Author**

1. Is the manuscript technically sound, and do the data support the conclusions?

Reviewer #1: Yes

Reviewer #2: Yes

2. Has the statistical analysis been performed appropriately and rigorously?

Reviewer #1: Yes

Reviewer #2: N/A

3. Have the authors made all data underlying the findings in their manuscript fully available?

Reviewer #1: Yes

Reviewer #2: Yes

4. Is the manuscript presented in an intelligible fashion and written in standard English?

Reviewer #1: Yes

Reviewer #2: Yes

Reviewer #1: Dear Author,

Thanks for your manuscript. It was a pleasure to read it as it is taughtfully wel written and conceptually well structured.

At some point I found it a bit resumed and with a feeling that this could even fit in three seperate studies a bit more structured and detailed.

As I was advancing in the review I missed an introduction to the moral domains and reflections through the studies how and why these moral domains had relatevely lower results then the Property and Equality domain. There was too much focus in these 2 moral domains throughout the study.

I liked that you tought carefully in your limitations about the cultural landscape, but I missed the same carefull thinking on the characteristics of your participants (e.g. gender. In study 1 it amounted to 83.3% of the sample. Does this influeced the result?)

Some minor reviews along the article:

Introduction:

- No refs. "Different days bring different moral tales..." -> ref here!

- "Still remains unclear..." -> To whom? From which starting point? Ref.

Morality And Social Life

- "Violating the relational..." - ref

Morality and Social Judgement

- "These findings ..." - Well written

- Good sub-chapter

Ethics and Data availability

- The first link is not well formatted. Remove the linking from the ')'

Method / Participants

- (Faul et al., 2009) - Ref bad formatted

Materials / Moral Domains

- "The current studies used a set of moral domains..." - Why?

- Table 1 - Bad formatted, Table 1 should appear right after the reference.

Study 2 / Participants

- "Cognition check" - which check?

Reviewer #2: Thank you for the opportunity to review the manuscript titled “Separate and unequal: Moral domains differ in corresponding social judgments of others” that was submitted to PLOS One. It deals with the question of whether different moral domains differ in their effect on social judgements and also examines differences in their effect on internal/external attributions and willingness to cooperate.

I find the idea raised by the authors thought-provoking and the studies generally well-executed. However, I have some concerns that make me wonder whether this research is informative enough on its own to warrant publication. Regardless of the outcome, I wish the authors all the best for their future research on this intriguing topic.

Major concerns

1. In the introduction, the authors state that there are distinct moral domains. However, they do not explain which frameworks exist in this regard and which one(s) they base their research on. More information regarding this is given in the Materials section of Study 1. However, since this information is relevant to all studies and important for understanding the basic idea behind the research, I suggest presenting it in the introduction section. Further information regarding the following questions should be added: Which are the most important frameworks of moral domains? Why were the two (Curry, Chesters, and Van Lissa’s MAC model and Rai and Fiske’s Relational Model) chosen? Why is the research based on both instead of choosing one? How is the theoretical idea underlying the present research (domains differing in their impact on social judgement) linked to the ideas presented in these frameworks? While the authors state that there is no research on whether the domains differ in their impact on social judgement, are there theoretically plausible ideas as to why some domains of the chosen frameworks might be more impactful than others? Adding this information would make a much stronger case for the theoretical idea and its translation into the methodology chosen by the authors.

2. Furthermore, I am missing some more information on why attributions were chosen as an additional dependent variable. This should be explained in the introduction as a basis for understanding the goal of the present research.

3. When the authors describe in the Materials section of Study 1 how the domains were chosen, it is not clear to me whether this was based on a separate pilot study. This should be made explicit. Furthermore, the authors used a bottom-up approach to choose their moral domains (i.e., having participants generate behaviors pertaining to the domains from the two frameworks from Curry et al. and Rai & Fiske and subsequently choosing and omitting some of the domains based on other participants’ ratings of these behaviors and potential overlaps). Why is this approach more informative for the author’s research question than choosing one of the frameworks and creating behaviors for all domains that are part of this framework? This would have enabled comprehensively examining one full framework instead of having an incomplete picture of both. An explanation of this should be given in the respective section of Study 1 and the potential drawbacks of picking some but not all domains from each of the frameworks (e.g., different scope of the domains from the two frameworks, risk of not covering all potentially relevant moral domains) should be discussed as a limitation in the General Discussion.

4. While the findings show that there are consistent differences in the effect of the chosen domains on social judgements, the authors did not investigate why some of the domains might be more important for social judgements than others, and the use of some but not all domains from two different frameworks makes it more difficult to deduct potential explanations. As the research is exploratory, the authors do not provide any theoretical explanation as to why some domains should be more impactful than others, either. In the General Discussion, the authors discuss this; yet, the research they cite does not provide a convincing explanation as to why some domains should be *more* impactful than others. They merely emphasize that the Equality and Property domains should be relevant for judgements that people make. An additional study investigating why some domains are more impactful than others could make this research much more informative. I leave it to the Editor to decide whether this is a necessary addition to the present research. If this is not further empirically investigated, I would be interested in a more detailed explanation in the General Discussion as to why the Equality and Property domains might be more impactful than the others.

Minor concerns

1. According to the study materials for all three studies, an additional two items were assessed together with the attribution items that were reported in the manuscript. Were these originally part of the attribution measurement and left out post hoc (if so why) or were they used to measure a different construct?

2. The main text states that that difference scores were calculated for the correspondent inferences. Yet, the y-axis in the graphs is titled Estimated Marginal Means and does not include any negative values.

3. A different term was used for the Equality/Fairness dimension in the main text compared to the graphs. Please consistently use one term.

4. While the preregistration for Study 3 states that there will be “two between-subjects conditions: one condition where cognitive load is increased by presenting moral stimuli alongside a span task, or one with just the moral stimuli presented alone”, both groups in Study 3 were confronted with a digit memorization task. This deviation should be acknowledged in the manuscript and the reason(s) for the deviation should be explained.

5. There seems to be a mistake in the following sentence: “We suspected that correspondent inferences, or inferences based on others’ behavior [16], made following domain-specific behaviors might highlight variations in how heavily these domains are considered when judging others” (p. 4, lines 79-81). Please correct it or otherwise make the sentence easier to understand.

**Do you want your identity to be public for this peer review?** For information about this choice, including consent withdrawal, please see our Privacy Policy

Reviewer #1: **Yes: ** Pedro Miguel Viegas Fernandes

Reviewer #2: No

---

## [Author Response · Author response to Decision Letter 1]

15 Aug 2025

I have included the body of our response to reviewers below, which includes a point-by-point response to specific reviewer and editor comments. A copy of this is also available in the submission files, titled "Response to Reviewers". We thank both the editor and the reviewers for their roles in peer reviewing our manuscript and appreciate the comments and feedback that were left. Please see our detailed response below:

Johannes Schwabe

Academic Editor

PLOS ONE

15 Aug 2025

Dear Dr. Schwabe,

On behalf of myself and my co-author, I am writing to resubmit our manuscript, “Separate and Unequal: Moral Domains Differ in Corresponding Judgments of Others” (PONE-D-25-18147), to PLOS One.

We greatly appreciate the time you and the reviewers took to provide detailed feedback. Using these comments, we have made revisions to the manuscript that I believe have enhanced its overall quality, with specific attention to adding clarity around our measures and the overall scientific contribution of the work. We have also revised the Supporting Information in line with your requests and those of the reviewers.

The file “Revised Manuscript with Track Changes” shows all changes that were made to the manuscript, which address most feedback that was given. The manuscript also includes an updated link to the primary data and analysis scripts, which have been updated to reflect the addition of gender analyses and minor corrections to the variable names. This link is currently a peer review link but will be made public if the manuscript were to be accepted to PLOS One.

Our point-by-point responses to editor and reviewer comments can be seen below, with the original comments included as “Comment:” and our response included as “Response:”.

Responses to the Editor:

Editor Comment: Theoretical Foundation: I agree with both reviewers that the theoretical foundation for your work needs improvement, and this applies to the very grounding of your work all across to final discussion. It is okay to not derive specific hypotheses, but your work does not exist in theoretical vacuum either. Relational models theory, moral foundations theory, and even research on moral identity can be drawn upon in the introduction and speaks to why certain domains might be more or less important in judging others moral character. For example, Haidt speaks about how violations in certain domains can disproportionately undermine global judgments because they signal "core flaws" (e.g., dishonesty). Also consider the work done on differential judgments of deontological versus utilitarian wrong- and right-doers (e.g., Jim Everett's work on this). Your research is currently presented without much of this context, which makes it hard for readers to understand the significance of your work and how it speaks to other findings.

Response: Thank you for pointing out the lack of clarity surrounding the theoretical foundation of this work. We included more theory in the introduction of our manuscript, paying special attention to exploring various models of moral domains and discussing these in the context of our research. We specifically added references to Everett and Haidt and spent more time developing the foundation for where our investigation into moral domains begins.

Editor Comment: I agree with reviewer 2 that the way you arrive at the domains studied needs to be more transparent. The methods underlying the domain selection are not clear enough, and we should learn a bit more about why two and why exactly those two frameworks were chosen to begin with.

Response: Thank you for pointing out this lack of transparency. We added more explanation of our creation of moral stimuli and why we feature a set of domains in our work that may feel potentially limited. Our methods and procedure have been made more transparent.

Editor Comment: The authors report the supplementary analysis as evidence that "the patterns cannot account for the results of Study 1". I tend to disagree. There are clear differences, and I feel like these results underline a central problem. Just because the positive and negative variants of a behavior create more polarized ratings on moral characters, why can we infer that the associated domain is more central for moral character judgments? Especially in combination with the fact that the behaviors were not sampled from a representative pool of all behavior potentially representing each domain, but arrived at in a fairly untransparent bottom-up approach, I do not see how we can really infer between-domain comparisons. This needs to be addressed more comprehensively.

Response: We agree that the supplemental information for this manuscript was lacking in an elaboration of included variables and why and how they are relevant to the patterns of results in our studies. We have done a complete overhaul of our Supporting Information, including a more in-depth description of the variables included in our supplementary analyses and explanations of those results in the context of our Study 1 results (S1 Text). This, in tandem with providing more explanation of our moral stimuli, should help to illustrate that the differences we found among domains within our studies cannot be explained by differences in features of the domains themselves.

Editor Comment: The lack of theorizing and embedding the research in the literature leads to a few issues. For example, in the pre-text to Study 2, the hypothesis appears, that there would be differences in how domain-specific behaviors influence participants' willingness to cooperate with social target. This is apparently purely data-driven, but there is a wealth of literature that would speak to that topic. This also relates to the findings how positive behaviors were perceived as driven more by targets' dispositions than negative behaviors. This conflicts with other studies consistently reporting that violations are given more weight in character judgments. Such apparent contradictions need to be transparent and discussed comprehensively.

Response: We have added more discussion surrounding the finding that positive behaviors were perceived as driven more by targets’ dispositions than negative behaviors. This finding may be attributable to the nature of our stimuli and asking participants to judge others on the basis of a singular behavior for each domain and valence, but in addition to this speculation we also included more theory-based reasoning for this finding.

Editor Comment: The power analysis of Study 1 is not well-founded. At this point, it is neither clear which exact effect you are interested in, which effect size would be reasonable to expect, neither which statistical test you are going to apply to test it. If this was just a convenience sample with no clear pre-specified hypothesis and power analysis, please just state that. If not, please describe the underlying assumptions more clearly.

Response: Thank you for pointing this out. We have added more clarity surrounding the design that was used in our initial power calculation.

Editor Comment: The use of difference scores and the exact statistical method planned should be explained in the methods section, and not in the results.

Response: We moved this discussion of difference scores to its own subsection within the Methods.

Editor Comment: The figures are not self-explanatory. The direction of the differences is not clear.

Response: We have revised each included figure using bars where there were previously lines, and added clearer figure labels. We have also revised the figure captions to make the direction of effects in each instance clearer.

Editor Comment: Please report tables detailing the Bonferroni-Tests in the supplementary material and refer to them when talking about their results.

Response: Our updated Supporting Information includes Bonferroni-adjusted multiple comparisons for each of our analyses to more clearly illustrate differences across our variables (S2, S4, and S6 Appendices). References to these have also been added to the main text where relevant.

Editor Comment: When comparing means, report the means and their SD in-text, not just the mean difference and CIs.

Response: Where mean differences and CIs were reported in the text, this has been replaced with M and SD.

Editor Comment: Generally, please report the individual variables (positive and negative, actor versus situation attribution, etc.) in the supplementary materials, and potentially even replace the current bar charts by dumb bell plots or similar, depicting both individual values as well as the differences.

Response: Dumbbell plots of our elaborated analyses not using difference scores have been added to the Supporting Information and referenced in the manuscript where relevant (S1, S3, and S5 Appendices).

Editor Comment: A clearer rationale for why we need the cognitive load paradigm is important would make it easier for readers to understand how Study 3 adds to the picture.

Response: More discussion of our cognitive load paradigm and its relevance to impressions and social judgment has been added to the manuscript.

Editor Comment: Please include standard means and correlations tables for all Studies, where possible based on individual variables, and add to the supplementary material if too clunky for the main manuscript.

Response: Standard means tables and correlations tables for each study are included at the end of the Supporting Information (S7 Appendix).

Responses to Reviewers:

Reviewer Comment: As I was advancing in the review I missed an introduction to the moral domains and reflections through the studies how and why these moral domains had relatevely lower results then the Property and Equality domain. There was too much focus in these 2 moral domains throughout the study.

Response: In addition to elaborating on our domains and stimuli creation, we added separate Discussion sections for each study to address findings incrementally.

Reviewer Comment: I liked that you tought carefully in your limitations about the cultural landscape, but I missed the same carefull thinking on the characteristics of your participants (e.g. gender. In study 1 it amounted to 83.3% of the sample. Does this influeced the result?)

Response: We have added gender analyses for each study (and the corresponding analysis scripts) to investigate whether the gender composition of our samples may have influenced our results. There was no indication that it did, which we have mentioned for each study in the manuscript.

Reviewer Comment:

- No refs. "Different days bring different moral tales..." -> ref here!

- "Still remains unclear..." -> To whom? From which starting point? Ref.

Morality And Social Life

- "Violating the relational..." - ref

Response: While these statements may have come off as referencing specific bodies of work, the first two statements are abstract musings of the phenomenon discussed in the manuscript and the gap it addresses. The third statement is continuing from a thought in the preceding sentence in which a relevant reference is included.

Reviewer Comment: Ethics and Data availability

- The first link is not well formatted. Remove the linking from the ')'

Response: Thank you for pointing this out. The links have been revised to link to an OSF repository for the purpose of peer review and may be made publicly available if the manuscript is accepted.

Reviewer comment: Method / Participants

- (Faul et al., 2009) - Ref bad formatted

Response: Thank you for pointing this out. This reference has been updated.

Reviewer comment: Materials / Moral Domains

- "The current studies used a set of moral domains..." - Why?

Response: Thank you for addressing the lack of clarity surrounding the domains. We have addressed this by adding more elaboration surrounding our domain stimuli and their creation.

Reviewer comment: Table 1 - Bad formatted, Table 1 should appear right after the reference.

Response: Thank you for catching this; this table has been moved up in the manuscript to appear after it is referred to in text.

Reviewer comment: Study 2 / Participants

- "Cognition check" - which check?

Response: I apologize if this was unclear; the manuscript has been updated to reflect that the same cognition check described in Study 1 was also used in the subsequent studies.

Reviewer Comment: In the introduction, the authors state that there are distinct moral domains. However, they do not explain which frameworks exist in this regard and which one(s) they base their research on. More information regarding this is given in the Materials section of Study 1. However, since this information is relevant to all studies and important for understanding the basic idea behind the research, I suggest presenting it in the introduction section. Further information regarding the following questions should be added: Which are the most important frameworks of moral domains? Why were the two (Curry, Chesters, and Van Lissa’s MAC model and Rai and Fiske’s Relational Model) chosen? Why is the research based on both instead of choosing one? How is the theoretical idea underlying the present research (domains differing in their impact on social judgement) linked to the ideas presented in these frameworks? While the authors state that there is no research on whether the domains differ in their impact on social judgement, are there theoretically plausible ideas as to why some domains of the chosen frameworks might be more impactful than others? Adding this information would make a much stronger case for the theoretical idea and its translation into the methodology chosen by the authors.

Response: Thank you for this comment. We have added more theory about the existence and development of moral domains to the Introduction section and have also included more detail about our specific domains and stimuli creation in the Materials section for Study 1. This should provide greater overall clarity regarding how our domains were selected and why, as well as how existing frameworks informed our stimuli.

Reviewer comment: Furthermore, I am missing some more information on why attributions were chosen as an additional dependent variable. This should be explained in the introduction as a basis for understanding the goal of the present research.

Response: Thank you for pointing out that the addition of attribution as an outcome variable was unclear. We have added an explicit explanation to the Introduction that discusses attribution in the context of relevant literature and why it was chosen as a variable.

Reviewer comment: When the authors describe in the Materials section of Study 1 how the domains were chosen, it is not clear to me whether this was based on a separate pilot study. This should be made explicit. Furthermore, the authors used a bottom-up approach to choose their moral domains (i.e., having participants generate behaviors pertaining to the domains from the two frameworks from Curry et al. and Rai & Fiske and subsequently choosing and omitting some of the domains based on other participants’ ratings of these behaviors and potential overlaps). Why is this approach more informative for the author’s research question than choosing one of the frameworks and creating behaviors for all domains that are part of this framework? This would have enabled comprehensively examining one full framework instead of having an incomplete picture of both. An explanation of this should be given in the respective section of Study 1 and the potential drawbacks of picking some but not all domains from each of the frameworks (e.g., different scope of the domains from the two frameworks, risk of not covering all potentially relevant moral domains) should be discussed as a limitation in the General Discussion.

Response: We have updated the manuscript to reflect that the creation of our stimuli was done in a series of pilot studies. We hope that the addition of more transparency surrounding our stimuli creation process manages to shed light on the reasoning for our creation process and that our intention was to increase the ecological validity of our stimuli, not just to limit our domains to a select few. However, we

---

## [Decision Letter · Decision Letter 1]

23 Sep 2025

Dear Dr. Adams,

Thank you for submitting your revised manuscript to PLOS ONE. I was able to receive a review from one of the original reviewers. The reviewer was pleased with your changes in response to the reviewer comments. However, they still identified a few questions they would like to see addressed in a final version of the paper. Therefore, we invite you to submit a revised version of the manuscript that addresses the points raised during the review process.

We look forward to receiving your revised manuscript.

Kind regards,

Corey Cook

Academic Editor

PLOS ONE

Journal Requirements:

Reviewers' comments:

Reviewer's Responses to Questions

**Comments to the Author**

Reviewer #2: (No Response)

2. Is the manuscript technically sound, and do the data support the conclusions?

Reviewer #2: Yes

3. Has the statistical analysis been performed appropriately and rigorously?

Reviewer #2: Yes

4. Have the authors made all data underlying the findings in their manuscript fully available?

Reviewer #2: Yes

5. Is the manuscript presented in an intelligible fashion and written in standard English?

Reviewer #2: Yes

Reviewer #2: Thank you for giving me the opportunity to review this revised version of the manuscript “Separate and unequal: Moral domains differ in corresponding social judgments of others”.

Overall, the authors have improved the manuscript. The reworking of the introduction and parts of the study descriptions have made the manuscript much easier to understand. Nevertheless, there are two points that have not been sufficiently addressed.

Before raising my remaining concerns, I would like to address one general point: For future revisions, please structure the response letter more clearly: Indicate which points were made by which reviewer and number each of the points made by the reviewers. For any changes you made in the manuscript, please indicate the page numbers and directly quote the changed text. Without doing so, reviewing your changes is very time-consuming and I cannot say with certainty that I have identified each of the (potentially multiple) passages you changed in response to a specific comment.

Remaining concerns:

1. The authors now mention that attributions are an additional DV in the introduction section. Apart from mentioning that this should play a role in person perception, however, they do not explain how this is linked to their theoretical idea. Why should different moral domains elicit different attributions? Given that attributions are one of the main DVs, this should be given more weight in the introduction.

2. Regarding your discussion of the different effects of the domains: So far, you have not added a suggestion as to why some domains should be *more* impactful than others. As you rightly point out on p. 26, “differences in how these moral behaviors manifest and their functions within social communities may influence how they are used in social impressions.” Yet, in your previous explanations on pp. 25f., you do not contrast the different manifestations or functions of the domains/behaviors corresponding to domains. You simply point out why the Property and Equality domains are relevant to daily life. However, the interesting factor would be: What differentiates them from the other domains? While you cannot answer this question based on your research and much of your research is exploratory to begin with, it would be interesting to at least derive some assumptions on the differences between the domains in the discussion.

**Do you want your identity to be public for this peer review?** For information about this choice, including consent withdrawal, please see our Privacy Policy

Reviewer #2: No

---

## [Author Response · Author response to Decision Letter 2]

31 Oct 2025

Below I have copied the Response to Reviewers, which is also included in the submission files as "Response to Reviewers."

Corey Cook

Academic Editor

PLOS ONE

27 Oct 2025

Dear Dr. Cook,

On behalf of myself and my co-author, I am writing to resubmit our manuscript, “Separate and Unequal: Moral Domains Differ in Corresponding Judgments of Others” (PONE-D-25-18147), to PLOS One.

We greatly appreciate the time you and the reviewers took to provide detailed feedback. We have used these comments to make revisions that have added more clarity to the manuscript and increased its quality. In particular, we have added more clarity surrounding our reasoning for including certain outcome measures and have more fully developed our discussion of our findings.

The file “Revised Manuscript with Track Changes” shows all changes that were made to the manuscript, which address the feedback from you as well as the feedback from Reviewer 2. Below I have included a point-by-point response to these comments with specific references to changes that have been made in the text where appropriate. In line with journal requirements, all citations have been checked for retractions. There is one additional citation included from the last version of the manuscript, which I have also disclosed below.

Our point-by-point responses to editor and reviewer comments can be seen below, with the original comments included as “Comment:” and our response included as “Response:”.

Responses to the Editor:

Editor Comment: If the reviewer comments include a recommendation to cite specific previously published works, please review and evaluate these publications to determine whether they are relevant and should be cited. There is no requirement to cite these works unless the editor has indicated otherwise.

Response: Thank you for bringing these requirements to our attention. The citations have been checked, and no known retractions have been issued for any of the papers referenced. In the revision process I added an additional citation to the paper to address comments from Reviewer 2. Surrounding citations have been changed to reflect the difference in number order of citations and the reference was added to the reference list.

Reviewer 2 Comment: For future revisions, please structure the response letter more clearly: Indicate which points were made by which reviewer and number each of the points made by the reviewers. For any changes you made in the manuscript, please indicate the page numbers and directly quote the changed text. Without doing so, reviewing your changes is very time-consuming and I cannot say with certainty that I have identified each of the (potentially multiple) passages you changed in response to a specific comment.

Response: Thank you for your comment. I have attempted to structure this response letter more clearly and include associated page numbers and quoted text where applicable.

Reviewer 2 Comment: The authors now mention that attributions are an additional DV in the introduction section. Apart from mentioning that this should play a role in person perception, however, they do not explain how this is linked to their theoretical idea. Why should different moral domains elicit different attributions? Given that attributions are one of the main DVs, this should be given more weight in the introduction.

Response: Thank you for this feedback; we have more fully developed the discussion of attribution as a DV by including why asymmetry in attribution across domains might matter to our research question. This is included on page 5 lines 99-109.

“Research has suggested that whether an individual perceives someone else’s behavior to be driven by their character or a situational circumstance may play a role in intuitive moral character judgments and shape beliefs about the kind of person someone is [22] [23]. In particular, dispositionally attributed behaviors are likely to be perceived as in line with someone’s personal beliefs and identity. We aimed to measure whether acts relevant to some moral domains may be more likely to be attributed to one’s character than acts in other domains. Some asymmetry in attribution following immoral acts has been suggested in past research, [24], though this research has primarily focused on how individuals may differently perceive moral norm violations that harm others vs. those that violate norms of purity. In our research, we aimed to investigate potential asymmetry between a larger set of proposed moral domains suggested by contemporary literature without focusing specifically on moral violations.”

Reviewer 2 comment: Regarding your discussion of the different effects of the domains: So far, you have not added a suggestion as to why some domains should be *more* impactful than others. As you rightly point out on p. 26, “differences in how these moral behaviors manifest and their functions within social communities may influence how they are used in social impressions.” Yet, in your previous explanations on pp. 25f., you do not contrast the different manifestations or functions of the domains/behaviors corresponding to domains. You simply point out why the Property and Equality domains are relevant to daily life. However, the interesting factor would be: What differentiates them from the other domains? While you cannot answer this question based on your research and much of your research is exploratory to begin with, it would be interesting to at least derive some assumptions on the differences between the domains in the discussion.

Response: This is a good point and we have further developed our discussion to speculate more about why these domains might differ from others. As you point out, we are unable to draw specific comparisons between the domains from our current research, but we have included some elaborated discussion on this topic on page 26 line 553-563.

“While our current findings are insufficient for making concrete differentiations between each of the domains, these different literatures help paint a picture of Equality- and Property-relevant behaviors as related to early socialization and embedded within social systems of development and everyday living. Within an American context, it is possible that these behaviors have evolved to be more ubiquitous and consistently valued than behaviors in other domains, which may be in line with moral research suggesting cultural variation in moral values based on differences in social ecologies and traditions when implementing or teaching moral beliefs [17]. Indeed, while moral behaviors may overall be adaptive solutions to problems of cooperation and relationship regulation within communities [8] [9], differences in how these moral behaviors manifest and their functions within social communities may influence how they are used in social impressions.”

Thank you again for your careful attention to our manuscript and considerate feedback. Both authors have read and approved of the revised manuscript and hope that it is suitable for publication in PLOS One.

Sincerely,

Savannah Adams

Department of Psychology

University of Michigan

Email: snladams@umich.edu

---

## [Editor Report · Decision Letter 2]

17 Nov 2025

Separate and unequal: Moral domains differ in corresponding social judgments of others

PONE-D-25-18147R2

Dear Dr. Adams,

We’re pleased to inform you that your manuscript has been judged scientifically suitable for publication and will be formally accepted for publication once it meets all outstanding technical requirements.

Kind regards,

Dr. Corey Cook

Academic Editor

PLOS ONE

---

## [Editor Report · Acceptance letter]

PONE-D-25-18147R2

PLOS One

Dear Dr. Adams,

I'm pleased to inform you that your manuscript has been deemed suitable for publication in PLOS One. Congratulations! Your manuscript is now being handed over to our production team.

Kind regards,

on behalf of

Dr. Corey Cook

Academic Editor

PLOS One